# Effect of Polydeoxyribonucleotide (PDRN) Treatment on Corneal Wound Healing in Zebrafish (*Danio rerio*)

**DOI:** 10.3390/ijms232113525

**Published:** 2022-11-04

**Authors:** Shan Lakmal Edirisinghe, Chamilani Nikapitiya, S. H. S. Dananjaya, Jungho Park, Dukgyu Kim, Dongrack Choi, Mahanama De Zoysa

**Affiliations:** 1College of Veterinary Medicine and Research Institute of Veterinary Medicine, Chungnam National University, Daejeon 34134, Korea; 2Zerone Bio Inc., 322-1 Sanhak Building, Dankook University, 119 Dandae-ro, Cheonan-si 31116, Korea

**Keywords:** cornea, epithelium, fibroblasts, injury, PDRN, wound healing, zebrafish

## Abstract

This study aimed to develop a corneal epithelial injury model in zebrafish (*Danio rerio*) and investigate the effectiveness of polydeoxyribonucleotide (PDRN) treatment on in vivo corneal epithelial regeneration and wound healing. Chemical injury to zebrafish cornea was produced by placing a small cotton swab containing 3% acetic acid solution. PDRN treatment was performed by immersing corneal-injured zebrafish in water containing PDRN (2 mg/mL) for 10 min at 0, 24, 48, and 72 h post-injury (hpi). The level of corneal healing was evaluated by fluorescein staining, histological examination, transcriptional profiling, and immunoblotting techniques. Fluorescein staining results demonstrate that PDRN treatment significantly (*p* < 0.05) reduced the wounded area of the zebrafish eye at 48 and 72 hpi, suggesting that PDRN may accelerate the corneal re-epithelialization. Histopathological evaluation revealed that injured corneal epithelial cells were re-organized at 72 hpi upon PDRN treatment with increased goblet cell density and size. Moreover, transcriptional analysis results demonstrate that PDRN treatment induced the mRNA expression of *adora2ab* (6.3-fold), *pax6a* (7.8-fold), *pax6b* (29.3-fold), *klf4* (7.3-fold), and *muc2.1* (5.0-fold) after the first treatment. Besides, *tnf-α* (2.0-fold) and heat-shock proteins (*hsp70*; 2.8-fold and *hsp90ab1*; 1.6-fold) have modulated the gene expression following the PDRN treatment. Immunoblotting results convincingly confirmed the modulation of Mmp-9, Hsp70, and Tnf-α expression levels upon PDRN treatment. Overall, our corneal injury model in zebrafish allows for understanding the morphological and molecular events of corneal epithelial healing, and ophthalmic responses for PDRN treatment following acid injury in zebrafish.

## 1. Introduction

The cornea is the anterior, transparent, and avascular outermost structure of the eye that serves an important role in vision (refractive), mechanical integrity (protective), and immunological defense functions [1,2]. It allows light to enter the eye and permits an external image to be displayed on to the retina. The human cornea comprises six layers, namely from exterior to anterior direction, epithelium, Bowman’s layer, stroma, basement membrane, Descemet’s membrane, and endothelium [3]. Furthermore, stratified squamous epithelium and stroma represents 80–85% of the total corneal thickness and it consists of highly organized collagen fibrils, proteoglycans, and keratocytes/fibroblasts [3,4]. The corneal collagen fibrils are the primary structural component of the corneal matrix, which maintains mechanical strength and barrier properties.

Due to external position, the cornea is extremely susceptible to a wide variety of injuries. Corneal damage, which leads to visual impairment and blindness, could be due to many conditions such as infections, ulceration, disorders, inflammation, and malnutrition [5]. Physiologically, the cornea exhibits series of complex wound healing responses to restore corneal homeostasis, transparency, integrity, and normal corneal anatomy. In contrast, the cornea can elicit long-term deleterious effects due to hyperactivity of inflammatory cytokines, growth factors, and chemokines under pathological conditions of wound healing [6]. These may affect infection, negative interaction of the corneal epithelial–stroma, abnormal extracellular matrix (ECM), loss of integrity, and stromal components [6,7,8,9]. Furthermore, the cornea is an excellent model to study the wound healing process because of its accessibility, less vascularization, and simple anatomical structure. Formerly developed corneal injury models were related with outermost superficial epithelial layer or deeper stromal destruction to study the healing of both epithelial and stromal layers [6,10].

Over the last two decades, mammalian animal models have supported a detailed understanding of the mechanisms involved in corneal wound healing [11,12,13,14,15]. According to the published literature, acid- or alkali-based corneal injuries have been reported using rabbit and rodent models [16,17,18,19]. Even though mammalian models exert beneficial outcomes for ophthalmological studies, several limitations such as ethical concerns, time-consuming, and cost of maintenance are considered as drawbacks when performing clinical studies [20,21]. Alternatively, the zebrafish has emerged as an excellent experimental model animal for human diseases [22,23,24,25,26,27]. So far, extensive embryological and mutagenesis approaches in zebrafish have been carried out for the retina, however, little is known about the zebrafish cornea and its regenerative capabilities against corneal injuries. Similar to humans, zebrafish cornea is highly specialized in terms of anatomical, molecular, and cellular functions [27,28,29]. Moreover, the fish cornea has highly specialized structural and physiological adaptations to tolerate a diverse range of aquatic habitats [30]. Several reports demonstrated the regeneration and repair mechanism of corneal endothelium followed by the surgical injury in zebrafish [31,32]. Zebrafish as modeling for corneal wound healing for ophthalmic inventions is still poorly described. Therefore, it is necessary to develop a relatively convenient, efficient, and low-cost animal corneal injury model and validation.

In this study, we used the PDRN developed from DNA isolated from salmon (*Salmon salar*) sperm cells. The structure of PDRN is described as a linear polymer, which forms the monomeric unit of pyrimidine nucleotides combining the purine and phosphodiester bonds. In humans, PDRN has been applied for various biomedical applications including wound healing and regeneration. Numerous studies demonstrated the role of PDRN on cell migration, ECM assembling, angiogenesis, and anti-inflammatory functions. Therefore, it is important to understand the pharmaceutical potentials and therapeutic applications of PDRN in different animal models [33,34,35]. However, the treatment capability of PDRN on ophthalmic applications has not yet been described using zebrafish. Therefore, we explored the PDRN effect on in vitro wound healing (HDFs), in vivo corneal epithelium regeneration, and wound healing by histological, transcriptional, and protein expression analysis. Notably, an alternative conformation by fluorescein staining was optimized during the epithelial regeneration and healing following acetic acid injury in the cornea of zebrafish eye.

## 2. Results

### 2.1. Cytotoxicity and Cell Proliferative Activity of PDRN

To determine the optimum treatment dose of PDRN on human dermal fibroblasts (HDFs), cytotoxicity and cell proliferation profiles were evaluated using 3-(4,5-Dimethylthiazol-2-yl)-2, 5-diphenyltetrazolium bromide (MTT), and Trypan blue dye exclusion assays, respectively. PDRN-treated (50–500 µg/mL) HDFs did not exhibit considerable toxic effects, which were indicated by over 50% of cell viability at 12, 24, and 48 h (Appendix A). Besides, cell viability was significantly (*p* < 0.05) higher in PDRN-treated HDFs at 150–250 µg/mL (24 and 48 h). Subsequently, cell viability was reduced dramatically when increasing the PDRN up to 500 µg/mL. PDRN-induced cell proliferation was highest at 200 µg/mL, and it gradually decreased by increasing the PDRN treatment up to 500 µg/mL. Based on cytotoxicity and cell proliferation results, PDRN doses of 100 and 200 µg/mL were selected for the in vitro wound healing assay.

### 2.2. Effect of PDRN on In Vitro Wound Healing in HDFs

The in vitro wound healing effect of PDRN was tested by analyzing the collective cell migration upon the creation of a pseudo-wound in the confluent layer of HDFs (Scratch assay). Quantification of the cell migration and the speed of wound closure were analyzed by measuring the open wound area (cell-free gap) upon PDRN treatment at 0, 18, and 36 h. The result illustrates that PDRN treatment (100 and 200 µg/mL) enhanced the HDFs migration in a concentration dependant manner, and decreased the cell-free gap (Figure 1A). The relative open wound area% was significantly (*p* < 0.001) reduced in PDRN-treated HDFs at 18 h (65 and 57% of 100 and 200 µg/mL, respectively) and 36 h (20 and 8% of 100 and 200 µg/mL, respectively) compared to that of the control (82 and 54% at 18 and 36 h, respectively) (Figure 1B). In particular, the almost complete closure of the cell-free gap was clearly visible at 36 h upon PDRN treatment (200 μg/mL) and the low-serum media (positive control).

### 2.3. Effect of PDRN on Corneal Wound Healing upon Acid Injury in Zebrafish

We determined the ability of the PDRN to regenerate or re-epithelialize the corneal epithelium following chemical (3% acetic acid) injury in an in vivo zebrafish model. Corneal fluorescein staining was used to assess the viability of injured epithelial layers (Figure 2A). The fluorescein sodium salt is widely used as a fluorescent tracer to distinguish the injured and un-injured corneal surfaces (Figure 2B). The lightly injured or eroded sites are indicated with individually scattered fluorescein, while severe injuries display as an area of confluent staining over the corneal surface. The quantitative assessment results determined the average wounded area (0.7 ± 0.06 mm^2^) by acetic acid treatment at 1 hpi. Immediately after corneal injury (0 hpi), no internal bleeding or visible defects were observed in eyes under microscopic observation. Over time, the fluorescence level was decreased in PDRN-treated (2 mg/mL) eyes at 24, 48, and 72 hpi (Figure 2C), indicating the accelerated re-epithelialization process. Especially, a sharp reduction of fluorescence level was evidenced in the PDRN-treated group after 48 hpi. We next performed the quantitative comparison of the fluorescence intensity and the results revealed a sharp increase of fluorescence level at 24 hpi in both groups, however, the PDRN-treated group had slightly lower intensity. Furthermore, the PDRN-treated group showed a significant (*p* < 0.05) reduction of fluorescence level (61.1 and 36.0%) compared to that of the vehicle-treated group (104.5 and 71.0%) at 48 and 72 hpi, respectively (Figure 2D). Overall, these results confirmed that PDRN treatment enhanced corneal healing following the acetic acid injury in zebrafish.

### 2.4. Histological Analysis of Corneal Re-Epithelialization in Acid-Injured PDRN-Treated Zebrafish

Hematoxylin and eosin (H&E) staining results displayed the characteristic corneal architecture with four to five epithelial cell layers (thickness 22 ± 1.7 µm) in an un-injured eye (Figure 3A). It consists of morphologically different three types of cells namely, basal (inner), intermediate, and mature superficial cells. The superficial cells were identified with a round nuclei, and a flat and elongated squamous morphology. Intermediate cell morphology was not unique and they are densely populated in the corneal epithelium. H&E stained sections of the cornea-injured eye contains an irregular shape cornea at 1 hpi, which confirmed the optimum conditions such as concentration (3%) and time (10 s) for acetic acid injury (Figure 3B). In contrast, a smooth and symmetrical shape corneal surface was noticeable in the un-injured eye of the negative control group (Figure 3A). Moreover, under higher magnification, it was clearly shown the complete loss of epithelium layers in the cornea injury group at 1 hpi (Figure 3C). In response to the PDRN treatment, re-epithelialization was accelerated by forming a neoepithelial basal cell layer at the central cornea during the initial restoration phase (data not shown). Initially, elongated basal cells deposits on the basement membrane and they appeared with uneven distribution. However, the microscopic results illustrated a more dispersed stroma, less cell integrity, and differentiation of the basal layer, which were not completed even at 24 hpi. This is further indicated by the higher fluorescein staining at 24 hpi of both injured groups (vehicle and PDRN) than that of the 1 hpi. The re-epithelialization process was further enhanced in the PDRN-treated group at 72 hpi, as shown by significant increase of (*p* < 0.01) epithelial thickness (18.0 ± 1.7 µm) compared to that of the cornea-injured vehicle-treated group (10.2 ± 1.3 µm) at 72 hpi (Figure 3D). Furthermore, microscopic observation confirmed the significant (*p* < 0.05) increase of the numbers of average cell layers in the cornea-injured PDRN-treated (average: 4 layers) and cornea-injured vehicle-treated (average: 2 layers) groups at 72 hpi (Figure 3E).

### 2.5. Effect of PDRN on Goblet Cell Characteristics in Corneal Wound Healing in Zebrafish

Mucins produced by goblet cells play a vital role in protecting the ocular surface therefore, we tested the hypothesis that a PDRN solution could accelerate the healing of corneal epithelium in zebrafish via modulating goblet cell density or the size. Periodic Acid-Schiff (PAS) staining results revealed the goblet cell distribution in three main regions of the cornea, namely ventral peripheral, central, and dorsal peripheral (Figure 4A). Relatively higher goblet cell numbers were observed at ventral and dorsal peripheral sites. The histological examination of the whole eye of injured groups (vehicle- and PDRN-treated) revealed a regular shape and multiple layers of regenerating corneal epithelium on the stroma at 72 hpi (Figure 4B). The magnified images of each group described the abundance of goblet cells in three regions of cornea (Figure 4C). Briefly, goblet cell density was the highest in dorsal and ventral peripheral sites with greater size distribution compared to that of the central region. Notably, the PDRN-treated group showed mature (larger) goblet cells compared to the vehicle-treated group and control at 72 hpi. Significantly (*p* < 0.05) higher goblet cell density was shown in PDRN-treated group (23.8 ± 1.4 cells/mm), compared to cornea-injured vehicle-treated group (12.5 ± 3.1 cells/mm). Meanwhile, the un-injured negative control group had the highest goblet cell density of 28.8 ± 2.6 cells/mm, at 72 hpi (Figure 4D). Additionally, we evaluated the effect of PDRN on the modulating surface area of goblet cells, and the results revealed that PDRN treatment significantly (*p* < 0.05) increased the surface area by 2.40-fold compared to the cornea-injured vehicle-treated group (Figure 4E). Overall, the increase of goblet cell density and surface area by PDRN treatment may enhance the production and secretion of mucins for lubricating the ocular surface to protect the cornea injured by acetic acid exposure.

### 2.6. Analysis of mRNA Expression during PDRN Mediated Corneal Wound Healing in Zebrafish

Our next aim was to compare the mRNA expression profiles obtained from eye samples of un-injured, cornea-injured vehicle-treated, and cornea-injured PDRN-treated zebrafish by quantitative real-time PCR (Figure 5). The selected genes belong to adenosine receptor (AR) family (*adora2a.1*, *adora2ab*, *adora1b*, and *adora2b*) and specific signaling cascades under wound healing and regeneration functions. The above genes are further sub-divided based on their specific functions such as inflammatory (*tnf-α*), tissue remodeling (*mmp9* and *mmp13*), proliferation (*tgfβ1*), ocular morphogenesis (*pax6a* and *pax6b*), protein folding (*hsp70* and *hsp90ab.1*) and goblet cell maturation, and mucin production (*klf4*, *muc2.1*, *muc5.1*, and *muc5.2*). The expression fold-change of the target gene in either cornea-injured vehicle-treated or cornea-injured PDRN-treated groups was calculated relative to the expression level of the un-injured control group.

#### 2.6.1. Expression Profiles of Adenosine Receptors (*adora2a.1*, *adora2ab*, *adora1b*, and *adora2b*)

It has been demonstrated that activation of the adenosine receptor pathway improves the tissue remodeling and anti-inflammatory responses. Based on the previous results on activation of A_2A_ by PDRN, we investigated the mRNA expression of four members of the adenosine receptor family in cornea-injured PDRN-treated eye. Among the selected genes, *adora2a.1* mRNA expression-fold was significantly (*p* < 0.05) increased in the PDRN-treated group (2.9- and 3.1-fold) compared to the vehicle-treated group (1.1- and 0.4-fold) at 24 and 48 hpi, respectively. In addition, *adora2ab* mRNA expression was significantly increased (6.3-fold; *p* < 0.001) in the PDRN group at 1 hpi compared to the expression (1.6-fold) in the vehicle-treated group. At the early stage of corneal healing (1 hpi), the transcription level of *adora1b* remained in the basal for PDRN- and vehicle-treated groups compared to that of the control group. However, the expression was significantly (*p* < 0.05) increased in the PDRN-treated group (2.4-fold) at 24 hpi compared to that of the vehicle group (0.8-fold). The expression of *adora2ab* mRNA was slightly (1.8-fold) induced with PDRN treatment at 24 hpi than the vehicle-treated control group (0.8-fold). Moreover, mRNA expression level of *adora2b* was only increased in vehicle-treated group (1.5-fold; *p* < 0.05) at 1 hpi compared to the PDRN group (0.6-fold). Except for the expression of *adora2a.1* in PDRN-treated group, all the other A_2A_ receptors did not considerably change their expression at 72 hpi in the control and vehicle-treated groups. Upon the PDRN treatment, the expression of A_2A_ receptors (*adora2a.1* and *adora2ab*) were the most responsive receptors compared to the other ARs during the first hour of corneal healing. Although not conclusive, our data suggest that PDRN modulates the members of adenosine receptor family in the zebrafish eye following corneal injury and PDRN treatment.

#### 2.6.2. Expression Profile of *tnf-α*

We examined the expression profile of *tnf-α* since it is associated with corneal inflammation, chemotaxis, immune response, angiogenesis, and wound healing. The cornea-injured vehicle-treated fish eye showed high transcriptional level of *tnf-α* at 1 (3.7-fold) and 24 hpi (5.8-fold), and those were lowered significantly (*p* < 0.05) in the injured PDRN-treated group, by 1.7- and 3.1-fold, respectively. The expression level of *tnf-α* was decreased below the basal level in the injured groups (cornea-injured vehicle- and cornea-injured PDRN-treated groups) at 48 hpi. Based on the results, we suggest that the PDRN effect on enhanced corneal epithelial wound healing may be partially mediated by modulating the inflammatory responses of *tnf-α*.

#### 2.6.3. Expression Profiles of *mmp9* and *mmp13*

The corneal epithelial cells secrete matrix metalloproteinases (MMPs) and overexpression and increased activity of MMPs may delay the remodeling or cause the degradation of ECM. Therefore, we examined whether PDRN affects Mmp-9 and Mmp-13 in cornea-injured zebrafish eyes. After the cornea injury, the *mmp9* level was strongly elevated in all the analyzed time points in corneal-injured vehicle-treated group and its peak expression (13.9-fold) was shown at 48 hpi. However, the corneal-injured PDRN-treated group had a relatively moderate upregulation of 2.0-fold (1 hpi), 2.8-fold (24 hpi), and 6.0-fold (48 hpi). Similar to the expression of *mmp9*, *mmp13* mRNA was higher in the corneal-injured vehicle-treated group at all selected time points. The peak expression of *mmp13* was at 24 hpi (20.6-fold). Moreover, the corneal-injured PDRN-treated group exhibited lower expression-fold of *mmp13* than that of the vehicle-treated group at 24, 48, and 72 hpi. Based on the results, we suggest that PDRN treatment may maintain a moderate expression of *mmp9* and *mmp13* in the cornea-injured eye. Further study on PDRN-regulated MMPs and ECM turnover may help to develop effective therapeutic application to treat corneal injuries.

#### 2.6.4. Expression Profile of *tgfβ1*

TGFβ1 is one of the key regulators of corneal epithelial wound healing and it is known to be involved in different epithelial repair processes such as fibroblast proliferation and formation of ECM. Therefore, we compared the *tgfβ1* mRNA expression in the corneal-injured PDRN- and vehicle-treated eye. The expression of *tgfβ1* was slightly increased in eye tissues of both PDRN- and vehicle-treated groups compared to that of the control at 1 and 24 hpi. However, the level of *tgfβ1* was higher at 1 hpi in the vehicle (1.9-fold) compared to that of the PDRN-treated (1.3-fold) group. The time-course expression pattern of *tgfβ1* did not show a considerable induction in both vehicle- and PDRN-treated groups compared to the control group.

#### 2.6.5. Expression Profiles of *pax6a* and *pax6b*

The PAX6 gene encodes a transcription factor, essential for the development of the corneal epithelium, which is involved in the maintenance and wound healing. The mRNA expression of both *pax6a* and *pax6b* were significantly (*p* < 0.001) elevated at 1 hpi in the corneal-injured PDRN-treated group. In fact, *pax6a* showed an upregulation (7.8-fold; *p* < 0.001) in the PDRN-treated group compared to that of the vehicle-treated group (1.2-fold). Similarly, *pax6b* was also induced (29.5-fold; *p* < 0.001) in the PDRN-treated group compared to the vehicle-treated group (3.7-fold) at 1 hpi. Even though both *pax6a* and *pax6b* expressions were drastically reduced at 24 h, the expressions were higher in the PDRN group compared to that of vehicle-treated group. Thereafter, the expression of *pax6a* and *pax6b* remained close to the basal level at 48 and 72 hpi in both cornea-injured groups (PDRN- and vehicle-treated).

#### 2.6.6. Expression Profiles of Heat Shock Proteins (*hsp70* and *hsp90ab1*)

The heat shock proteins (HSPs) are molecular chaperones that have been documented to be involved in epithelial wound healing by directing intracellular transport and preventing pathologic aggregation. Here, the selected *hsp70* and *hsp90ab1* were drastically (*p* < 0.05) upregulated in the PDRN group at 1 hpi (17.5- and 2.5-fold, respectively) when compared to the corneal-injured vehicle-treated group. The highest expressions of *hsp70* and *hsp90ab.1* were observed in the vehicle-treated group at 1 hpi (47.2- and 5.7-fold, respectively). Thereafter, the expression of *hsp90ab1* was slightly induced in the vehicle-treated group at 48 and 72 h compared to that of the PDRN-treated group. The overall expression pattern showed that the vehicle-treated group had higher *hsp70* and *hsp90ab.1* mRNA expression pattern compared to that of the PDRN-treated group.

#### 2.6.7. Expression Profiles of *klf4*

KLF4 is a zinc finger transcription factor that maintains cellular homeostasis in diverse epithelial tissues including the cornea. The results illustrated that for both the PDRN- (7.3-fold) and vehicle-treated groups, the corneal *klf4* mRNA transcripts were significantly (*p* < 0.001) upregulated (6.8-fold) compared to that of the control at 1 hpi. Thereafter, the resulting *klf4* mRNA expression levels were downregulated (<1.0-fold) in both PDRN- and vehicle-treated groups.

#### 2.6.8. Expression Profiles of Mucins (muc2.1, muc5.1, and muc5.2)

Mucins secreated by goblet cells are essentail to maintain healthy corneal epithelial functions. Here, we analyzed the mRNA expression level of three isoforms of mucins (muc2.1, 5.1, and 5.2). The results showed that muc2.1 was significantly (*p* < 0.05) upregulated in both the PDRN (5.0-fold) and vehicle (4.3-fold) treated groups compared to that of the control at 1 hpi. Thereafter. muc2.1 expression was not considerably changed until 72 h among the groups. In contrast, muc5.1 and 5.2 were downregulated (<1.0-fold) in both the PDRN- and vehicle-treated groups at 1 hpi. The mRNA expressions of both muc5.1 and muc5.2 were slightly increased, and muc5.1 expression was significantly upregulated at 48 hpi in the PDRN-treated group compared to that of the vehicle. At 24 hpi, muc5.2 showed significant (*p* < 0.05) induction in the PDRN-treated group compared to the vehicle. The muc5.3 were not shown any considerable fold changes in both injured groups compared to that of the control (data not shown).

### 2.7. Immunoblot Analysis of the PDRN-Treated Zebrafish with Corneal Injury

To further evaluate the PDRN effect on corneal wound healing, we examined the protein expression of Mmp-9, Hsp70, and Tnf-α by immunoblot analysis. The Mmp-9 expression was relatively low in the un-injured control group and in the corneal-injured vehicle-treated eye at 1 and 24 hpi (Figure 6A). However, the protein level of Mmp-9 was slightly higher in the corneal-injured PDRN-treated group at 24 hpi (Figure 6B). The highest expression of Mmp-9 was in the corneal-injured vehicle-treated group at 48 hpi. Moreover, Mmp-9 expression was significantly higher in the vehicle-treated group (3.3- and 2.4-fold) compared to that of the PDRN group (1.2- and 0.8-fold) at 24 and 72 hpi, respectively. Based on the downregulation pattern of Mmp-9 protein level (at 48 and 72 hpi), it is possible to suggest that PDRN treatment may facilitate the tissue stability after initial wound healing. Next, protein expression of Hsp70 was significantly (*p* < 0.05) induced in the corneal-injured vehicle-treated group (2.8-fold) when compared to the control at 1 hpi, and remained higher than the PDRN-treated group until 48 hpi. The results illustrated that Hsp70 level was higher in the corneal-injured PDRN-treated group (1.9-fold) compared to the corneal-injured vehicle-treated group (0.8-fold) at 72 hpi.

During the early wound healing stage (1 hpi), an upregulated expression of Tnf-α was shown in the corneal-injured vehicle-treated (7.6-fold; *p* < 0.01) and PDRN-treated groups (4.3-fold) when compared to the control group. Interestingly, Tnf-α level was decreased in the corneal-injured PDRN-treated group with the time, which indicates the reduction of inflammatory status in corneal-injured eye.

## 3. Discussion

Zebrafish are proven to be a powerful experimental animal model for studying embryonic development, genetic screening, diseases, and assessment of drug efficacy and toxicity [22,23,24,25,26,36,37,38,39,40]. The fish have camera-type eyes, and their cornea structures have been found to be anatomically similar to mammals. Therefore, we selected zebrafish for developing a corneal injury model. On the other hand, PDRN has shown multiple biological activities associated with regeneration and wound healing, thus, we investigated the therapeutic capability of PDRN on corneal epithelial wound healing following the cornea injury. We found that PDRN was not cytotoxic to the HDFs, up to 250 µg/mL, and increases the HDFs live cells in a dose-dependent manner up to 200 µg/mL. These results are overlaid with the previous report, which described the viability and proliferation effects of PDRN on HDFs [41]. Accordingly, 100 and 200 µg/mL of PDRN doses were selected for the in vitro wound healing study. Results of enhanced cell migration activity in PDRN-treated HDFs provides strong evidence that PDRN could be applied for medicating different types of corneal injuries.

Firstly, we demonstrated the application of fluorescein sodium salt to evaluate the re-epithelialization of damaged cornea in zebrafish. Fluorescein sodium salt penetrates through the intercellular gaps of corneal epithelial cells due to the loss of tight junction/integrity [42,43,44]. We confirmed that PDRN treatment enhanced the rapid corneal healing with a high recovery rate of epithelium, which can be quantified by fluorescein staining. The increased fluorescence intensity at 24 hpi indicates the loss of epithelial integrity and increased apoptotic cells following the cornea injury. Upon corneal re-epithelialization, fluorescence intensity on the outer corneal surface was decreased considerably during the first three days. Besides, in the present method, fluorescence quantification was determined in the whole eye. In contrast, fluorescence quantification has been determined in rodent cornea [45,46]. Moreover, compared to the rodent models, the cornea of zebrafish exerts higher epithelial recovery in a short time gap, which would be beneficial for future studies related to corneal injury.

To investigate the effect of PDRN on the cornea re-epithelization process and goblet cell density or size, histological analysis was performed. Previous studies have demonstrated that the zebrafish and human cornea’s outermost epithelium has similar anatomical structures. In agreement with the above, our histological results showed that the non-keratinized, squamous cells consist of three morphologically different epithelial cells (superficial, intermediate, and basal), which represent approximately 60% of the total corneal thickness. Different cell morphology in the epithelium illustrates that the life cycle of the cells reaches from immature dividing basal cells over gradually maturing intermediate cells to the mature outermost superficial cells. Upon cornea injury, the outer epithelium was completely removed from the basal cells on the basement membrane with a slightly dispersed stroma of the zebrafish eye. A previous study reported that re-epithelialization occurs due to the simultaneous migration of proliferated cells from both dorsal and ventral peripheral areas to the corneal center [47,48]. Wijnholds et al. described that the corneal limbus represents both the peripheral of the cornea, which is required for the production of specialized basal columnar epithelial cells during the epithelium regeneration [49]. Thus, we assumed that even at the complete corneal epithelium damage in zebrafish, it has potential to activate epithelial regeneration and healing from peripherals to the central cornea, similar to stem cell activation in corneal wound healing in human.

In contrast to aquatic animals, terrestrial animals have more functional conjunctival goblet cells, which produce and secrete soluble mucins and maintain a tear film. Gipson et al. described the arrangement of conjunctival goblet cells and factors affecting their differentiation [50]. In humans, most of the conjunctival goblet cells appear singly in regions of sparse density and as clusters in the forniceal region [50]. Additionally, the distribution pattern of goblet cells within the corneal epithelium varies among species. Our data show that those corneal goblet cells are present in zebrafish and their niche on the corneal surface. Moreover, comparatively large and well-differentiated goblet cells were observed in both corneal peripherals, and the central cornea contains less density with smaller goblet cells. Similar to the mammalian models, goblet cells show a plum shape that is in contact with the superficial and intermediate cells, thus, the goblet cells have not extended the entire thickness to the basal membrane. However, the goblet cell density of the three regions was considerably less than the other rodent models. Based on above results, we suggest that PDRN promotes epithelial cell differentiation including the goblet cells, which may reflect the organization of secretory mucin through mature goblet cells. Upon the re-epithelialization, those goblet cells were formed on the new epithelium with a comparatively enlarged size than the goblet cells in un-injured cornea.

To understand the molecular mechanism of corneal epithelial wound healing in zebrafish upon PDRN treatment, transcriptional responses were analyzed. PDRN degradation generates active oligo-and mononucleotides, purines, and pyrimidines, which become available for biological activity. PDRN is known to act on the A_2A_ purinergic receptors to activate the downstream signaling cascade for cell growth and neogenesis [35,51,52]. Our data showed that among the tested four receptors, considerable mRNA expression was observed for *adora2a.1* and *adora2ab* within 48 hpi in the PDRN-treated group. Even though *adora1b* showed slight induction, both *adora1b* and *adora2b* may be relatively non-responsive receptors for PDRN treatment in zebrafish. Lowering of the *tnf-α* mRNA expression upon PDRN treatment was observed at the early stages of epithelial healing (1 hpi) compared to the vehicle-treated group. This result was confirmed through the similar lowering expression pattern of Tnf-α at the protein level at 1 hpi in response to the PDRN exposure. Previous studies also revealed the inhibition of corneal inflammation by topical PDRN treatment in experimental keratoconjunctivitis sicca in a rat model [33], and our data strongly supported to show anti-inflammatory properties of PDRN exposure in zebrafish. After the corneal injury, several MMPs are upregulated by transcription or activation in response to the cytokines [53]. MMPs are involved in cell migration via the degradation of ECM or by modifying cellular adhesive properties during corneal wound healing [54,55,56]. Especially in corneal injuries, excessive expression of MMPs may dissolve the epithelial basement membrane (BM) and ECM, which negatively affects corneal wound healing [57,58]. In this study, the mRNA expression of *mmp9* and *mmp13* was sharply increased in the corneal-injured vehicle-treated group compared to that of the PDRN-treated group at 1 to 72 hpi. However, *mmp9* expression of the PDRN-treated group considerably increased only at 48 hpi. The results were further confirmed by the level of Mmp-9 protein, which was increased even at 72 hpi in the corneal-injured vehicle-treated group compared to that of the PDRN-treated group. We suggest that elevation may be occurring in the remodeling stage, which is involved in epithelial cell movement by disrupting cell–cell contacts, as described in Mauris et al. [56]. Next, Mmp-9 was suppressed in response to the PDRN treatment (72 hpi), undergoing strong epithelial barrier formation and corneal epithelial healing. Mmp-13 is expressed only at the basal layer of the healing corneal epithelium [59,60], suggesting its involvement for repairing the basal membrane of the corneal epithelium. According to the expression pattern of Mmp-9 and Mmp-13, we assumed that the PDRN treatment might regulate the MMPs at desirable levels in epithelial cells and BM during the corneal healing process. Harber et al. described that Tgf-β1 inhibits the proliferation of cultured equine epithelial cells and keratocytes [61], but another study reported its stimulatory effects on corneal stromal fibroblast cell proliferation [62]. In this study, the *tgfβ1* mRNA expression pattern was slightly increased in both injured groups (PDRN and vehicle treatments), but was not considerably different. The slight increased level of *tgfβ1* (at 1 and 24 hpi) may be due to minor stromal disruption upon injury and PDRN can mediate the expression of Tgf-β1.

Koroma et al. described that Pax6 is involved in corneal maintenance and wound healing [63], but their effect on corneal development and wound healing was further observed by Pax6-overexpressing transgenic mouse models [64,65]. In zebrafish, *pax6a* and *pax6b* mRNA expression was induced in corneal-injured PDRN-treated group at the early stage of corneal wound healing (1 and 24 hpi). Our data demonstrate that *pax6* isoforms (*pax6a* and *pax6b*) in zebrafish could play a role in regeneration and wound healing similar to the mouse model.

The Klf4 involves in epithelial cell ablation, regeneration of epithelial barrier, and formation of conjunctival goblet cells [66,67]. We observed elevated *klf4* mRNA expression in the corneal-injured zebrafish (PDRN and vehicle) at 1 hpi compared to that of the un-injured (control) group. We assumed that this elevated *klf4* might be due to the epithelial disruption by cornea injury. In contrast, downregulated mucin (*muc5.1* and *5.2*) expression (except the *muc2.1*) might reduce the number or size of the conjunctival goblet cells at 1 hpi. We suggest that upon PDRN treatment, mucin (*muc5.1* and *5.2*) showed an increasing pattern of expression at the later stage during epithelial regeneration and differentiation of the conjunctival goblet cells. Together, *klf4* and other mucin expression patterns (*muc5.1* and *5.2*) clearly explained that PDRN accelerated the epithelial cell differentiation, which restored the mucin secretion.

Increased expression of Hsp70 has been demonstrated in actively migrating cells upon tissue injury or cellular degeneration [68]. Our data, supported by previous studies, show that both *hsp70* and *hsp90ab.1* mRNA expression increased sharply in both injured groups at 1 hpi, which facilitates the cellular degeneration of the initial stage of corneal healing. Similarly in other corneal models [68], Hsp70 expression in zebrafish was induced with PDRN treatment as a protective function against stress conditions experienced by epithelial cells. Furthermore, the level of *hsp90ab.1* was slightly expressed in the corneal-injured vehicle-group at the later stage of healing, whereas the PDRN-treated group showed fast epithelial regeneration. Moreover, the expression pattern of Hsp70 further validated the elevated *hsp70* mRNA expression in both injured groups at 1 hpi. These findings suggest that Hsp70 act as the most stress-responsive factor during the zebrafish corneal re-epithelilization process.

## 4. Materials and Methods

### 4.1. Cell Culture and Cytotoxicity of PDRN on HDFs

To determine the cytotoxicity and optimum dose of PDRN, MTT colorimetric assay was performed. In brief, adult primary HDFs (ATCC PCS-201-012, Manassas, VA, USA) were cultured in fibroblast basal medium (PCS-201-030, Manassas, VA, USA) supplemented with fibroblast growth kit-low serum (PCS-201-041, Manassas, VA, USA) at 37 °C in a 5% CO_2_ incubator with a humidified (85–95%) atmosphere. As described in our previous study, HDFs (2.5 × 10^5^ cells per well) were seeded in a 96-well microplate and incubated for 12 h with the same conditions as described previously [69]. After 12 h, the medium was replaced and HDFs were treated with different concentrations (50–500 µg/mL) of PDRN (Zerone Bio, Inc., Cheonan-si, Chungcheongnam-do, Korea), and cells were further incubated to measure time-course cell viability assessment at 12, 24, and 48 h. Following the incubation, the culture medium was replaced with a fresh medium, 10 µL of MTT (5 mg/mL; Sigma-Aldrich, St. Louis, MO, USA) was added to each well, and the plate was incubated at 37 °C for 4 h in the 5% CO_2_ incubator. The resulting formazan crystals were dissolved in 50 µL of dimethyl sulfoxide (DMSO) (Sigma-Aldrich, St. Louis, MO, USA), and we measured the absorbance at 570 nm with a microplate absorbance reader (iMark, Bio-Rad Laboratories, Inc., Hercules, CA, USA). The HDFs treated with medium alone were considered as 100% viable (control). The trypan blue assay was conducted together with cell viability assay to further confirm the proliferative activity of different PDRN exposure only at 24 h. The following day, HDFs were seeded in a 6-well plate (1.5 × 10^5^ cells per well) supplemented with serum-free fibroblast basal medium at 37 °C in 5% CO_2_. After 24 h incubation, two sets of cell samples were treated with different concentrations (50–500 μg/mL) of PDRN and kept in an incubator for 24 h. After the treatment, cell viability assay was conducted as aforementioned. For the proliferation assay, cells were collected using 0.25% trypsin–EDTA (Welgene, Gyeongsan-si, Gyeongsanbuk-do, Korea), pelleted, and resuspended in the same medium. The cells were then stained with an equal volume of 0.2% (*w*/*v*) trypan blue solution (Sigma-Aldrich, St.Louis, MO, USA), and live (colorless) and dead (stained blue) cells were counted with Luna-FL Dual Fluorescence Cell Counter (Logos Biosystems, Anyang-si, Gyeonggi-do, Korea). All the experiments were performed in triplicates.

### 4.2. In Vitro Wound Healing Activity of PDRN

In vitro wound healing activity was investigated upon PDRN exposure to HDFs according to our previously described method [69]. Briefly, collected HDFs suspension was seeded at a density of 2 × 10^5^ cells/mL into each well (70 μL) of a culture-insert 2 well (ibidi GmbH, Munich, Gräfelfing, Germany). After the 24 h incubation, the insert was gently removed to create a cell-free gap (500 μm), resembling wound creation. The dishes were filled with serum-free basal medium (2 mL) containing 100 and 200 μg/mL of PDRN. For control and positive control, sterile deionized water (200 µL), and fibroblast low serum (2% FBS) medium (2 mL) were used, respectively. Images of each dish were captured at 0, 18, and 36 h using an inverted light microscope (Leica^®^ DMi8, Wetzlar, Mannheim, Germany), to determine the time taken to fill the cell-free gap by cell migration. All experiments were performed in triplicate (*n* = 3) and cell-free gaps were measured using ImageJ software (ImageJ, ver. 1.6, Bethesda, MD, USA) by normalizing to the cell-free gap at 0 h.

### 4.3. Corneal Injury and Fluorescein Staining of Zebrafish Eye

Wild-type zebrafish were maintained in an automated water circulation system under a 12 h light:12 h dark photoperiod at 28 °C. All the fish were selected under similar bodyweight and other growth parameters to reduce the experimental bias. Selected fish (*n* = 108; mean weight 0.49 ± 0.08 g) were divided into three groups, namely: control, vehicle, and PDRN. Before the experiment, fish were acclimatized in 20 L tanks for one week. During the acclimatization, fish were fed twice a day with a commercial diet supplemented with artemia. All the experiments with zebrafish were conducted in accordance with the approved guidelines and regulations of the Animal Ethics Committee of Chungnam National University (202109-CNU-126).

For corneal injury, fish were anesthetized using 0.12% (*w*/*v*) of Tricaine (Sigma-Aldrich, St. Louis, MO, USA), and the central corneal epithelium of the right side eye was touched (10 s) using 3% acetic acid dipped pointed cotton swab (prepared using the cotton threads inserting into the 1 mL pipet tip). The injured fish were transferred to another recovery tank for 5 min. The injured cornea was treated with freshly prepared PDRN (pH 7.2), or vehicle control (nucleus-free water) at 0, 24, 48, and 72 h. The control consisted of uninjured fish that also followed a similar anesthetizing procedure to reduce the experimental bias. Corneal abrasions of individually maintained fish (*n* = 16) were visualized using sterile fluorescein sodium salt (Sigma-Aldrich, Saint Louis, MO, USA). Images were taken at 0, 24, 48, and 72 h post-injury (hpi) following fluorescein staining using a stereo microscope (Leica^®^ S8 APO, Wetzlar, Mannheim, Germany) equipped with a camera (Leica^®^ KL300 LED, Wetzlar, Mannheim, Germany) and Fluorescence adapter (NIGHTSEA, Lexington, MA, USA). The total eye areas were quantitated for fluorescence intensity using ImageJ software (NIH, ver. 1.6), and we graphed the measured intensity by normalizing to 0 h of each group. The normalized results were reported at each time point as intensity% during corneal wound healing.

### 4.4. Analysis of the Effect of PDRN on Corneal Healing by Histological Analysis

Zebrafish right eye was collected for histology at 0, 24, 48, and 72 h after PDRN treatment, and were immediately placed in Hartmann fixation (Sigma-Aldrich, St. Louis, MO, USA) for 20 min before being transferred to 70% ethanol. Tissues were dehydrated by passing through an ascending graded series of alcohol and cleared with xylene in a Semi-enclosed Benchtop Tissue Processor (Leica^®^ TP1020, Wetzlar, Mannheim, Germany). Thereafter, eye tissues were embedded in paraffin wax (Leica^®^ EG1150 Tissue Embedding Center, Wetzlar, Mannheim, Germany) and sectioned into 3 μm thickness (Leica^®^ RM2125 microtome, Wetzlar, Mannheim, Germany) for H&E (ab245880, Abcam, Cambridge, Cambridgeshire UK) or PAS (ab150680, Abcam, Cambridge, Cambridgeshire UK) staining according to the manufacturer’s protocols. Finally, stained tissue slides were imaged using a digital camera (LEICA^®^ DCF450-C, Wetzlar, Mannheim, Germany) connected to a microscope (LEICA^®^ DM 3000 LED, Wetzlar, Mannheim, Germany).

### 4.5. Quantitative Real-Time PCR (qRT-PCR) Analysis

To determine the immune gene expression upon the PDRN treatment on corneal healing, qRT-PCR was performed. Right side eyes were collected from eight zebrafish in each group (un-injured, corneal-injured vehicle-treated, and corneal-injured PDRN-treated) at 1, 24, 48, and 72 hpi. All tissues were immediately snap-frozen in liquid nitrogen and stored at −80 °C until RNA isolation. Total RNAs were extracted using TRIzol reagent (Invitrogen, Waltham, MA, USA). RNA concentration was measured using NanoDrop One (Thermo Scientific, Waltham, MA, USA). Total RNA (1.25 µg) was used to synthesize the cDNA using Prime script 1st strand cDNA synthesis kit (Takara^®^, Tokyo, Japan) according to the manufacturer’s instructions. The expression pattern of treated or untreated zebrafish corneal wound healing related key genes were examined by qRT-PCR analysis. Briefly, adenosine receptors (*adora1b*, *adora2a.1*, *adora2ab*, and *adora2b*) with corneal wound healing genes, such as tumor necrosis factor alpha (*tnf-α*), matrix metalloproteinase −9 and −13 (*mmp9* and *mmp13*), transforming growth factor-beta1 (*tgfβ1*), paired box -6a and -6b (*pax6a* and *pax6b*), kruppel-like factor 4 (*klf4*), mucins (*muc2.1*, *5.1*, and *5.3*), and heat shock proteins (*hsp70* and *hsp90ab1*) were selected, and the respective gene-specific primers were designed as shown in Appendix A. The relative mRNA expression level was determined using the 2^−ΔΔCt^ method [70].

### 4.6. Analysis of PDRN Effect on Wound Healing Associated Genes

Immunoblot analysis was performed for isolated eye tissues from control, vehicle, and PDRN-treated zebrafish at 0, 24, 48, and 72 hpi. The isolated tissues were homogenized (1 min) with 300 μL of ice-cold lysis buffer, (pH 7.6) (ProEXTM CETi, TransLab Inc., Daejeon, Chungcheongnam-do, Korea). The respective homogenates were centrifuged at 12,000 rpm for 10 min at 4 °C. Bradford protein assay (Bio-Rad Laboratories, Inc., Hercules, CA, USA) was used to quantify the protein levels in each tissue sample. Samples were then denatured with 2X Laemmli sample buffer (Sigma Aldrich, St Louis, MO, USA), at 100 °C, for 5 min, and an equal amount of protein (35 μg) was loaded onto 10% Sodium Dodecyl Sulfate-Poly Acrylamide Gel Electrophoresis (SDS-PAGE) and electrophoresed at 80 V for 30 min, and subsequently at 110 V for 3 h. The proteins were trans-blotted onto Polyvinylidene Difluoride (PVDF) membranes for 2 h using a Trans-Blot semidry transfer cell (Bio-Rad Laboratories, Inc., Hercules, CA, USA), and then blocked with 5% bovine serum albumin (BSA). The membranes were incubated with Mmp-9 (#ARP3390-T1100; AVIVA Systems Biology, San Diego, CA, USA), Hsp70 (#4872; Cell Signaling Technology, Danvers, MA, USA), Tnf-α (#KP1540Z-100; Kingfisher Biotech, Saint Paul, MN, USA) antibodies at 1:1000 dilutions, or β-actin specific monoclonal antibodies (sc47778; Santa Cruz Biotechnology, Dallas, TX, USA) at 1:3000 dilution in 5% BSA overnight at 4 °C. Thereafter, the membranes were washed thrice with Tris-buffered saline containing Tween 20 (TBST) and incubated for another 1 h with HRP-labeled secondary antibodies at 1:3000 dilution Anti-mouse IgG (gtx213111-01; GeneTex, Irvine, CA, USA) or anti-rabbit IgG (7074 s; Cell Signaling Technology, Danvers, MA, USA) in 5% BSA at 25 °C. The specific proteins ware carried out using a chemiluminescence detection system (Fusion Solo S, Vilber, Lourmat, France). The protein bands intensities were quantified using the Evolution-CAPT software (FUSION software user and service manual-v17.03, Vilber Company, San Sebastiàn, Spain), and normalized with respect to the expression of β-actin to obtain the relative protein expression fold. Three independent experiments were carried out to quantify the average expression of Mmp-9, Hsp70, and Tnf-α.

### 4.7. Statistical Analysis

All analyses were performed using GraphPad Prism software version 5 for Windows (GraphPad Software Inc., San Diego, CA, USA). Data were analyzed using one-way and/or two-way analysis of variance (ANOVA) to determine the overall significance between the experimental groups and/or time points when necessary. Moreover, Bonferroni’s post hoc test and/or Student’s two-tailed *t*-test were conducted to compare the average means of control and treatment. Significant difference was considered at *p* < 0.05.

## 5. Conclusions

In conclusion, we developed a fluorescein staining technique together with a histological analysis to evaluate the acid-based corneal injury and epithelial wound healing in zebrafish eye. Primarily, PDRN-treated HDFs display rapid cell migration and wound healing activities in vitro. Most importantly, PDRN can promote the epithelial wound healing in the acid-based corneal injury model in zebrafish by activating the rapid cell proliferation, migration, increasing the goblet cells, anti-inflammatory responses, and modulation of genes belonging to AR family, regeneration, and wound healing functions. Overall, our corneal injury model in zebrafish allows for a better understanding of the morphological and molecular events of corneal epithelial healing, and the ophthalmic responses for PDRN treatment following acid injury in zebrafish.

## Figures and Tables

**Figure 1 ijms-23-13525-f001:**
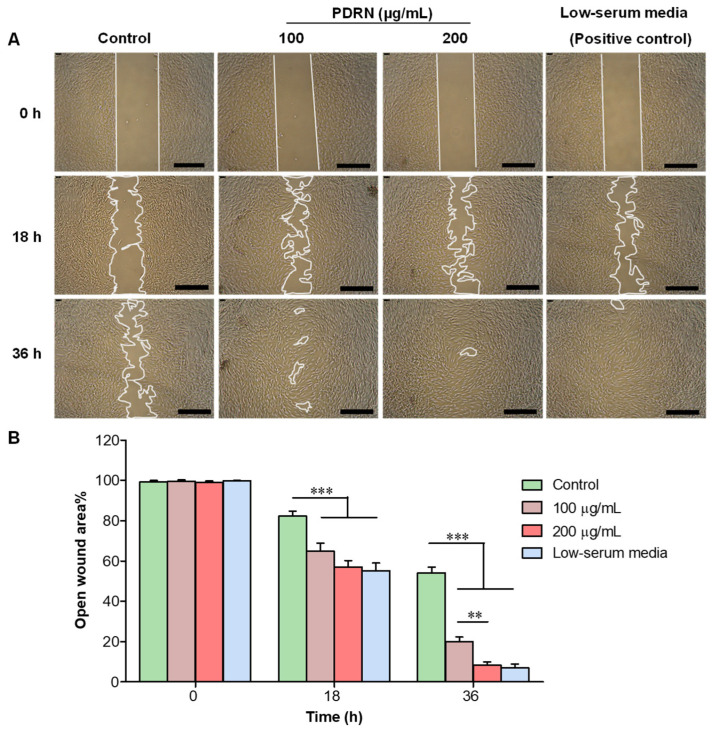
Effect of PDRN on in vitro wound healing in HDFs. (**A**) Representative images of in vitro scratch assay, and (**B**) quantification of cell-free area of HDFs upon PDRN treatment (100 and 200 µg/mL) at 0, 18, and 36 h. Results are presented as means ± standard deviation (SD) based on the time point at 0 h (Two-way ANOVA, ** *p* < 0.01, *** *p* < 0.001, Scale bar 500 µm, *n* = 3).

**Figure 2 ijms-23-13525-f002:**
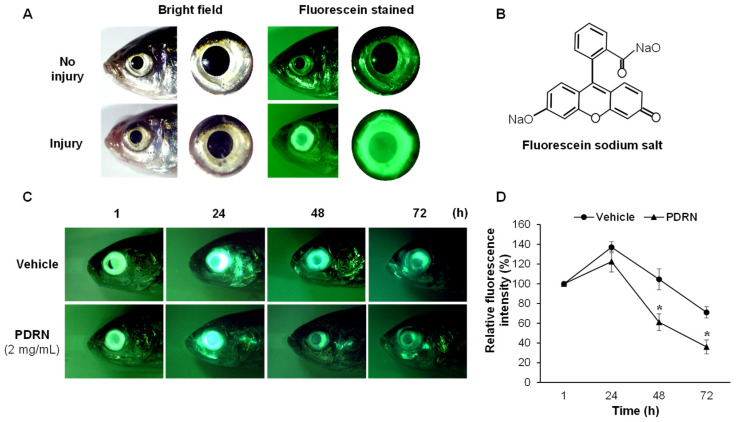
Examination of PDRN effect on corneal re-epithelialization upon acetic acid injury. (**A**) The representative photographs of fluorescein-stained zebrafish eye of the control and the acid injury groups. (**B**) The chemical structure of fluorescein sodium salt. (**C**) The time-dependent fluorescence intensity change at the injury site of the eye and (**D**) quantification of the relative fluorescence intensity of cornea-injured PDRN- and vehicle-treated groups. Data are presented as the mean ± standard deviation (SD). Student’s two-tailed *t*-test was performed to find statistical significance, * *p* < 0.05 for the PDRN group vs. the vehicle group (*n* = 16).

**Figure 3 ijms-23-13525-f003:**
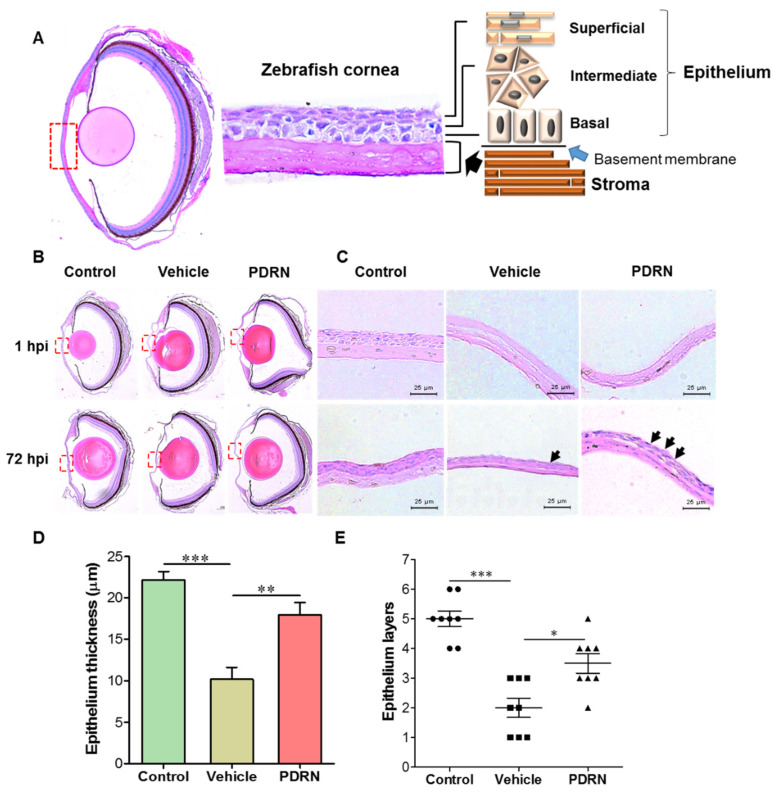
Histological analysis of zebrafish eye and corneal re-epithelialization following the corneal injury and PDRN treatment. (**A**) Representative image of H&E stained zebrafish eye, cornea, and graphical illustration of the corneal epithelium. (**B**) Representative photograph of the whole eye of un-injured (negative control), cornea-injured vehicle, and cornea-injured PDRN-treated groups (scale bars, 40 μm), and (**C**) photographs of selected central epithelium surface of the cornea (scale bars, 200 μm.) at 1 and 72 hpi. Regenerated epithelial cell layers were marked with black arrows. Quantification of corneal regeneration based on (**D**) epithelium thickness, and (**E**) number of epithelium cell layers at 72 hpi. One-way ANOVA was performed to determine the statistical significance; * *p* < 0.05, ** *p* < 0.01, *** *p* < 0.001 (*n* = 8).

**Figure 4 ijms-23-13525-f004:**
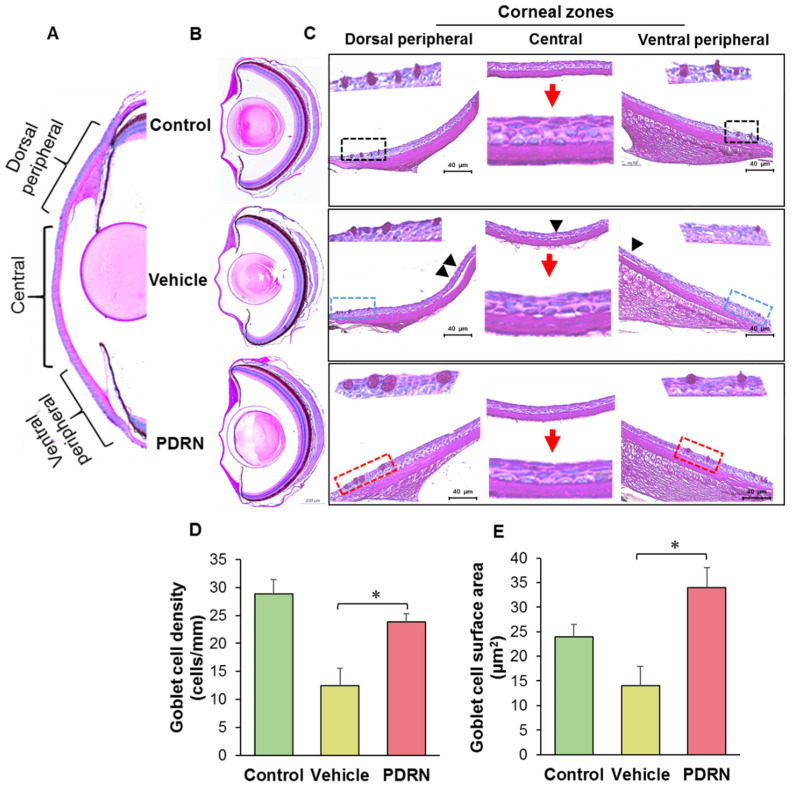
PDRN effect on goblet cell characteristics in the corneal epithelium following the cornea injury and PDRN treatment. (**A**) Magnified transverse sections stained with PAS showing the representative panoramic overview of three corneal epithelial zones (dorsal peripherals, central, and ventral peripheral) and matured goblet cells at 72 hpi after the cornea injury (Scale bars, 200 μm). (**B**) Images showing the mature goblet cells (red square) in the PDRN-treated group, and (**C**) magnified zones showing that the epithelium cells are not well attached (black arrowheads) in the cornea- injured vehicle-treated group. The graphs illustrate quantification of (**D**) goblet cell density and (**E**) the goblet cell surface area (an indicator of size) following the cornea injury at 72 hpi. Goblet cell density was measured as the number of cells/mm of corneal surface (one-way ANOVA was performed to find statistical significance; * *p* < 0.05, *n* = 8).

**Figure 5 ijms-23-13525-f005:**
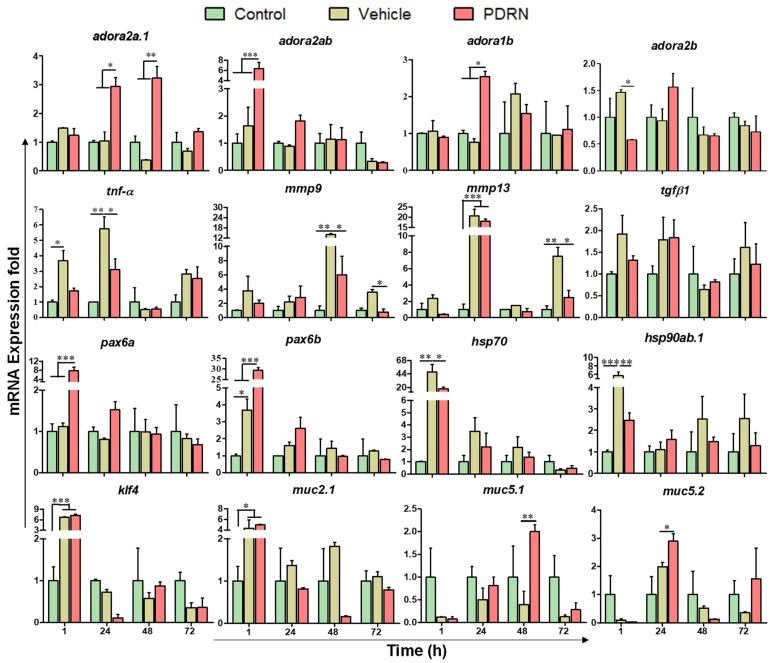
Transcriptional profiles of wound healing related genes in zebrafish eye upon cornea injury and PDRN treatment. The data are presented as fold changes over un-injured, cornea injury vehicle-treated, and cornea injury PDRN-treated. Three samples (R = 3) were collected from the right-side eye of nine adult fish for each condition, and two independent experiments were performed (two-way ANOVA followed by Dunnett’s post hoc test was performed to find statistical significance; * *p* < 0.05, ** *p* < 0.01, *** *p* < 0.001).

**Figure 6 ijms-23-13525-f006:**
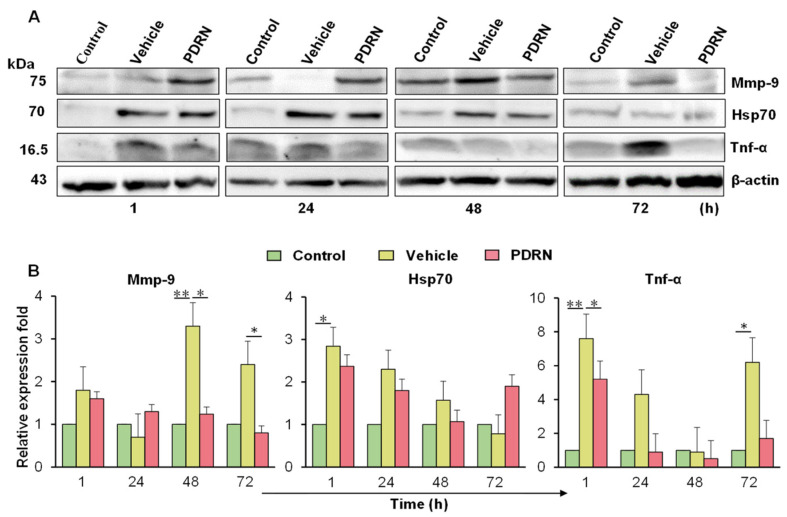
Immunoblot analysis of Mmp-9, Hsp70, and Tnf-α in response to corneal injury and PDRN treatment in zebrafish eye. (**A**) Western blot analysis images representing the specific protein bands that were expressed against the selected antibodies. (**B**) The pixel intensity of each protein band was quantified using Chemi Doc (Fusion Solo, Vilber, Lourmat, France) and normalized to β-actin. The relative expression fold was calculated based on the control group of each time point (one-way ANOVA was performed to find statistical significance; * *p* < 0.05; ** *p* < 0.01).

## Data Availability

The data presented in this study are available on request from the corresponding author.

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
