# Peer review of "Effect of Polydeoxyribonucleotide (PDRN) Treatment on Corneal Wound Healing in Zebrafish (Danio rerio)"

_ijms, 2022, doi:10.3390/ijms232113525_

Round 1

Reviewer 1 Report

In this paper, Edirisinghe et al. explored the effect of polydeoxyribonucleotide (PDRN) on corneal wound healing in zebrafish. The authors demonstrated that the zebrafish is a reliable model for studying corneal wound healing and the efficacy of PDRN in reducing inflammatory mediators and accelerating corneal epithelial differentiation and healing. The overall study design, methods, and the other sections of the manuscript, including figures, are well organized and illustrate the facts discussed in the text. The interpretations are consistent with the findings. However, I have some questions related to the experiments conducted in this paper. Below are my concerns:

  1. Supplementary figure S1: Authors should also perform time-course experiments (12h, 24h, 48h, and 72h) using different concentrations of PDRN for cell viability assay. 

  1. Figure 1: There is a mismatch between figure legends and what is represented in the figure. Figure legend shows time points as 0, 12, 24, and 36 h whereas, Figure 1B does not have all the time points. 

The manuscript has many typos and grammatical errors, I have listed some of them:

  1. Line 339> cornel wound>> corneal wound
  2. Line 382-384> In this study, quantified fluorescein intensity that created though the photographs showed a clear correlation between the PDRN treatments for fast corneal wound healing.>> Please rewrite this sentence.
  3. Line 385> showing ‘with’ the increased>> delete the word ‘with’
  4. Line 386-387>..was resulted…>> Please rewrite this sentence. 
  5. Line 449> may occurring>> should be written as ‘may occur’ or ‘may be occurring. 
  6. Line 474> resulted by>> resulted in

A more tightly worded manuscript with the above-suggested corrections will make the paper a good read.

Author Response

Please find the author’s responses to the referee’s comments attached separately.

Reviewer 2 Report

1.     In Figure 4C, the differences between Vehicle and PDRN groups should be compared instead of Vehicle and Control groups. Please modify.

2.     In line 582, “the respective gene-specific primers were designed as shown in Table 2.”; however, Table 2 was not found in the manuscript. Please add.

3.     In the section of “original-images”, the protein molecular weight ladder looks very unclear, and it is recommended to replace other prestained protein ladder in the future.

4.     In line 48, “cornea exhibit a complex wound”; it is suggested to change “exhibit” to “exhibits”.

5.     In line 100, 103 and 104, “in-vitro” please change to “in vitro”; usually, italics and no hyphens.

6.     In line 228, “mRNa” please change to “mRNA”.

7.     In line 232, “Transcription” please change to “transcription”; It's in the middle of the sentence.

8.     In line 301, there is two Spaces in front of “Thereafter”. Please modify.

Author Response

(The authors gave the same response as above.)

Round 2

Reviewer 2 Report

1.     In Figure 3D, the image was overstretched, please adjust.

2.     In Figure 3C and Figure 4C, please add scale bars on the images.

Author Response

Reviewer 2

1.Reviewer comment

In Figure 3D, the image was overstretched, please adjust.

Author’s comment

New figure was adjusted and replaced.

2.Reviewer comment

In Figure 3C and Figure 4C, please add scale bars on the images.

Author’s comment

Scale bars are included in the figures.
